# ESMValTool v2.0 – Technical overview

Mattia Righi[1], Bouwe Andela[2], Veronika Eyring[1,3], Axel Lauer[1], Valeriu Predoi[4], Manuel Schlund[1], Javier Vegas-Regidor[5], Lisa Bock[1], Björn Brötz[1], Lee de Mora[6], Faruk Diblen[2], Laura Dreyer[7], Niels Drost[2], Paul Earnshaw[7], Birgit Hassler[1], Nikolay Koldunov[8,9], Bill Little[7], Saskia Loosveldt Tomas[5], and Klaus Zimmermann[10]

[1]Deutsches Zentrum für Luft- und Raumfahrt (DLR), Institut für Physik der Atmosphäre, Oberpfaffenhofen, Germany
[2]Netherlands eScience Center, Science Park 140, 1098 XG Amsterdam, the Netherlands
[3]University of Bremen, Institute of Environmental Physics (IUP), Bremen, Germany
[4]National Centre for Atmospheric Science, University of Reading, Department of Meteorology, Reading, United Kingdom
[5]Barcelona Supercomputing Center, Barcelona, Spain
[6]Plymouth Marine Laboratory, Prospect Place, The Hoe, Plymouth, Devon, United Kingdom, PL1 3DH
[7]Met Office, FitzRoy Road, Exeter, EX1 3PB, United Kingdom
[8]Alfred Wegener Institute, Helmholtz Centre for Polar and Marine Research, Bremerhaven, Germany
[9]MARUM - Center for Marine Environmental Sciences, Bremen, Germany
[10]Swedish Meteorological and Hydrological Institute (SMHI), Norrköping, Sweden

*Correspondence to:* Mattia Righi (mattia.righi@dlr.de)

**Abstract.** This paper describes the second major release of the Earth System Model Evaluation Tool (ESMValTool), a community diagnostic and performance metrics tool for the evaluation of Earth System Models (ESMs) participating in the Coupled Model Intercomparison Project (CMIP). Compared to version 1.0, released in 2016, ESMValTool version 2.0 (v2.0) features a brand new design, with an improved interface and a revised preprocessor. It also features a significantly enhanced diagnostic part that is described in three companion papers. The new version of the ESMValTool has been specifically developed to target the increased data volume of CMIP Phase 6 (CMIP6) and the related challenges posed by the analysis and the evaluation of output from multiple high-resolution or complex ESMs. The new version takes advantage of state-of-the-art computational libraries and methods to deploy an efficient and user-friendly data processing. Common operations on the input data (such as regridding or computation of multi-model statistics) are centralized in a highly optimized preprocessor, which allows applying a series of preprocessing functions before diagnostics scripts are applied for in-depth scientific analysis of the model output. Performance tests conducted on a set of standard diagnostics show that the new version is faster than its predecessor by about a factor of three. The performance can be further improved, up to a factor of more than 30, when the newly-introduced task-based parallelization options are used, which enable the efficient exploitation of much larger computing infrastructures. ESMValTool v2.0 also includes a revised and simplified installation procedure, setting of user configurable options based on modern language formats, and high code quality standards following the best practices for software development.

# 1 Introduction

The future generations of ESM experiments will challenge the scientific community with an increasing amount of model results to be analyzed, evaluated and interpreted. The data volume produced by CMIP5 (Taylor et al., 2012) was already above 2 Petabytes and it is estimated to grow by about one order of magnitude in CMIP6 (Eyring et al., 2016a). This is due to the growing number of processes included in the participating models, the improved spatial and temporal resolutions, and the widening number of model experiments and participating model groups. Not only the larger volume of the output, but also the higher spatial and temporal resolution and complexity of the participating models is posing significant challenges for the data analysis. Besides these technical challenges, the variety of variables and scientific themes covered by the large number (currently 23) of CMIP6-endorsed Model Intercomparison Projects (MIPs) is also rapidly expanding.

To support the community in this big data challenge, the ESMValTool (Eyring et al., 2016c) has been developed to provide an open-source, standardized, community-based software package for the systematic, efficient and well documented analysis of ESM results. The ESMValTool provides a set of diagnostics and metrics scripts addressing various aspects of the Earth system that can be applied to a wide range of input data, including models from CMIP and other model intercomparison projects, and observations. The tool has been designed to facilitate routine tasks of model developers, model users, and model output users, who need to assess the robustness and confidence in the model results and evaluate the performance of models against observations or against predecessor versions of the same models. Version 1.0 of the ESMValTool was specifically designed to target CMIP5 models, but the growing amount of data being produced in CMIP6 motivated the development of an improved version, implementing a more efficient and systematic approach for the analysis of ESM output as soon as the output is published to the Earth System Grid Federation (ESGF, https://esgf.llnl.gov/), as also advocated in Eyring et al. (2016b).

This paper is the first in a series of four presenting ESMValTool v2.0 and it focuses on the technical aspects, highlights its new features and analyzes its numerical performance. The new diagnostics and the progress in scientific analyses implemented in ESMValTool v2.0 are discussed in the companion papers: Eyring et al. (2019), Lauer et al. (2019), and Weigel et al. (2019).

A major bottleneck of ESMValTool v1.0 (Eyring et al., 2016c) was the relatively inefficient preprocessing of the input data, leading to long computational times for running analyses and diagnostics, whenever a large data volume needed to be processed. A significant part of this preprocessing consists of common operations, such as time subsetting, format checking, regridding, masking, calculating temporal and spatial statistics, etc., which are performed on the input data before a specific scientific analysis is started. Ideally, these operations, collectively named preprocessing, should be centralized in the tool, in a dedicated preprocessor. This was not the case in ESMValTool v1.0, where only a few of these pre-processing operations were performed in such a centralized way, while most of them were applied within the individual diagnostic scripts. This resulted in several drawbacks, such as slow performance, code duplication, lack of consistency among the different approaches implemented at the diagnostic level, and unclear documentation.

To address this bottleneck, ESMValTool v2.0 has been developed: this new version implements a fully revised preprocessor addressing the above issues, resulting in dramatic improvements in the performance, as well as in the flexibility, applicability and user friendliness of the tool itself. The revised preprocessor is fully written in Python 3 and takes advantage of the data

abstraction features of the Iris library (Met Office, 2010-2019) to efficiently handle large volumes of data. In ESMValTool v2.0 the structure has been completely revised and now consists of an easy-to-install, well documented Python package providing the core functionalities (ESMValCore), and a set of diagnostic routines. The ESMValTool v2.0 workflow is controlled by a set of settings that the user provides via a configuration file and an ESMValTool recipe (called namelist in v1.0). Based on the

user settings, the ESMValCore reads in the input data (models and observations), applies the required preprocessing operations and writes the output to netCDF files. These preprocessed output files are then read by the diagnostics and further analyzed. Writing the preprocessed output to a file, instead of storing it in memory and directly passing it as an object to the diagnostic routines, is a requirement for the multi-language support of the diagnostic scripts. Multi-language support has always been one of the ESMValTool main strengths, to allow a wider community of users and developers with different level of programming

knowledge and experience to contribute to the development of the ESMValTool by providing innovative and original analysis methods. As in ESMValTool v1.0, the preprocessing is still performed on a per-variable and per-dataset basis, meaning that one netCDF file is generated for each variable and for each dataset. This follows the standard adopted by CMIP5 (and other MIPs), which requires that data for a given variable and model is stored in an individual file (or in a series of files covering only a part of the whole time-period in case of long time series).

To give ESMValTool users more control on the functionalities of the revised preprocessor, the ESMValTool recipe has been extended with more sections and entries. To this purpose, the YAML format (http://yaml.org/) has been chosen for the ESMValTool recipes and consistently for all other configuration files in v2.0. The advantages of YAML include an easier to read and more user-friendly code and the possibility for developers to directly translate YAML files into Python objects.

Moreover, significant improvements are introduced in this version for provenance and documentation: users are now pro-
vided with a comprehensive summary of all input data used by the ESMValTool for a given analysis and the output of each analysis is accompanied by detailed metadata (such as references and figure captions) and by a number of tags. These allow to sort the results by, e.g., scientific theme, plot type, or domain, thereby greatly facilitating collecting and reporting results, for example on browsable web-sites. Furthermore, a large part of the ESMValTool workflow manager and of the interface, handling the communication between the Python core and the multi-language diagnostic packages at a lower level, have been

completely rewritten following the most recent coding standards for code syntax, automated testing, and documentation. These quality standards are strictly applied to the ESMValCore package, while for the diagnostics more relaxed standards are used to allow a larger community to contribute code to the ESMValTool. As for v1.0, ESMValTool v2.0 is released under the Apache license. The source code of both ESMValTool and ESMValCore is freely accessible on the GitHub repository of the project (https://github.com/ESMValGroup) and is fully based on freely available packages and libraries.

This paper is structured as follows: the revised structure and workflow of ESMValTool v2.0 are described in Sect. 2. The main features of the new YAML-based recipe format are outlined in Sect. 3. Section 4 presents the functionalities of the revised preprocessor, describing each of the preprocessing operations in detail as well as the capability of the ESMValTool to fix known problems with data sets and to reformat data. Additional features, such as the handling of external observational datasets, provenance and tagging, as well as the automated testing are briefly summarized in Sect. 5. The progress in performance

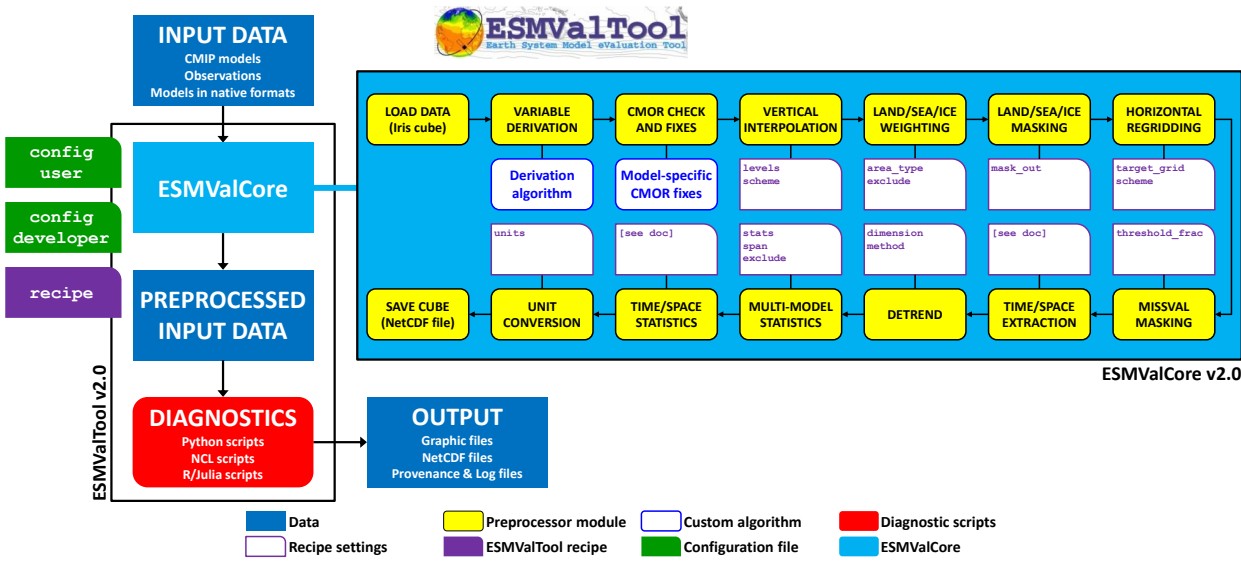

**Figure 1.** Schematic representation of ESMValTool v2.0.

achieved with this new version is analyzed in Sect. 6, where results of benchmark tests compared to the previous version are presented for one representative recipe. Section 7 closes with a summary.

This manuscript aims at providing a general, technical overview of ESMValTool v2.0. For more detailed instructions on the ESMValTool usage, users and developers are encouraged to take a look at the extensive ESMValTool documentation available
on Read the Docs (https://esmvaltool.readthedocs.io/).

## 2    Revised design, interface, and workflow

ESMValTool v2.0 has been completely redesigned to facilitate the development of core functionalities by the core development team, on the one hand, and the implementation of new scientific analyses (diagnostics and metrics) by diagnostic developers and the application of the tool by the casual users, on the other hand. These two target groups typically have different levels
of programming experience: highly experienced programmers and software engineers are maintaining and developing the core functionalities that affect the whole tool, while scientists or scientific programmers are mainly contributing new diagnostics and analyses to the tool. A schematic representation of ESMValTool v2.0 is given in Fig. 1.

ESMValTool v2.0 is distributed as an open-source package containing the diagnostic code and related interfaces, while the core functionalities are located in a Python package (ESMValCore), which is distributed via the Python package manager
or via Conda (https://www.anaconda.com/) and which is installed as a dependency of the ESMValTool during the installation procedure. The procedure itself has been greatly improved over v1.0 and allows installing the ESMValTool and its dependencies using Conda just following a few simple steps. No detailed knowledge of the ESMValCore is required by the users and scientific

developers to run the ESMValTool or to extend it with new analyses and diagnostic routines. The ESMValCore package is developed and maintained in a dedicated public GitHub repository, where everybody is welcome to report issues, request new features, or contribute new code with the help of the core development team. The ESMValCore can also be used as a stand-alone package, providing an efficient preprocessor that can be utilized as part of other analysis workflows or coupled with different software packages.

The ESMValCore contains a task manager that controls the workflow of the ESMValTool, a method to find and read input data, a fully revised preprocessor performing several common operations on the data (see Sect. 4), a message and provenance logger, and the configuration files. The ESMValCore is installed as a dependency of the ESMValTool and it is coded as a Python library (Python v3.7) which allows all preprocessor functions to be re-used by other software or used interactively, for example from a Jupyter Notebook (https://jupyter.org/). The new interface for configuring the preprocessing functions and the diagnostics scripts from the recipe is very flexible: it allows, for example, designing custom preprocessors (these are pipelines of configurable preprocessor functions acting on input data in a customizable order) and it allows each diagnostic script to define its own settings. The new recipe format also allows the ESMValTool to perform validation of recipes and settings, and to determine which parts of the processing can be executed in parallel, greatly reducing the run-time (see Sect. 6).

Although the ESMValCore is fully programmed in Python, multi-language support for the ESMValTool diagnostics is provided, to allow a wider community of scientists to contribute their analysis software to the tool. ESMValTool v2.0 supports diagnostics scripts in Python 3, NCL (NCAR Command Language, v6.6.2, https://www.ncl.ucar.edu/), R (v3.6.1, https://www.r-project.org/), and, since this version, Julia (v1.0.4, https://julialang.org/). Support for other freely available programming languages for the diagnostic scripts can be added on request. The coupling between the ESMValCore and the diagnostics is accomplished using temporary interface files generated at run-time for each variable-diagnostic combination. These files contains all the information that a diagnostic script may require to be run, such as the path to the preprocessed data, the list of input datasets, the variable metadata, the diagnostic settings from the recipe, the destination path for result files and plots, etc. The interface files are written by the ESMValCore preprocessor in the same language as the recipe (YAML, see Sect. 3), which highly simplifies the coupling. An exception is NCL which does not support YAML and for which a dedicated interface file structure has been introduced based on the NCL syntax.

ESMValTool v2.0 adopts modern standards for storing configuration files (YAML v1.2), data (netCDF4), and provenance information (W3C-PROV, using the Python package prov v1.5.3). Professional software development approaches such as code review (through GibHub pull requests), automated testing and software quality monitoring (static code analysis and a consistent coding style enforced through unit tests) ensure that the ESMValTool is reliable, well documented and easy to maintain. These quality control practices are enforced for the ESMValCore package. For the diagnostic code standards are somewhat more relaxed, since compliance to all of these standards can be quite challenging and may introduce unnecessary hurdles for scientists contributing their diagnostic code to the ESMValTool.

## 3 New recipe format

To allow a flexible and comprehensive user control on the many new features of ESMValTool v2.0, a new format for the configuration files defining datasets, preprocessing operations, and diagnostics (the so-called recipe) is introduced: YAML is used to improve user readability of the numerous settings and to facilitate their passing through to the diagnostics code, as well as the communication between the ESMValCore and the diagnostics.

An ESMValTool v2.0 recipe consists of four sections: documentation, preprocessors, datasets and diagnostics. Within each of these sections, settings are given as lists or as additional nested dictionaries for more complex settings. This allows controlling many options and features at different levels in an intuitive way. The diagnostic package contains many example recipes that can be used as a starting point to create more complex and extensive applications (see the companion papers for more details). In the following, each of the four sections of the recipe is described. A few ESMValTool recipes are provided in the Supplement as a reference.

### 3.1 Documentation section

This section of the recipe provides a short description of its content and purpose, together with the (list of) author(s) and project(s) which supported its development, and the corresponding reference(s). All these entries are specified using tags which are defined in the references configuration file (`config-references.yml`) of the ESMValTool. At run-time, the recipe documentation is collected by the provenance logger (see Sect. 5.2), which translates the tags into full text string and adds them to the output produced by the recipe itself.

### 3.2 Datasets section

This section replaces the `MODELS` section used in the ESMValTool v1.0 namelists and it is now called `datasets` to account for the fact that not only models, but also observations or reanalyses can be listed here. The datasets to be processed by all diagnostics in the recipe are provided as a list of dictionaries, each containing predefined sets of key-value pairs that unambiguously define the dataset itself. The required keys depend on the project class of the dataset (e.g. CMIP3, CMIP5, CMIP6, OBS, obs4mips, etc.) and are defined in the developer configuration file (`config-developer.yml`) of the tool. Typically, the user does not need to edit this file but only to provide the root path to the data in the user configuration file (`config-user.yml`). Based on the information contained in both files, the tool reconstructs the full path of the dataset(s) to locate the input file(s). During the ESMValTool execution, the dataset dictionary is always combined with the variable dictionary defined in the diagnostic section (see Sect. 3.4) into a single dictionary, such that the key-value pairs for the dataset and for the variable can be given in either dictionary. This has several advantages, for example the same dataset can be defined for multiple variables from different mips (such as the CMIP5 "Amon" and "Omon"), just defining the common keys in the dataset dictionary and the variable-specific one (e.g., `mip`) in the variables dictionaries. The tool collects the dataset information by combining the keys from the two dictionaries, depending on the variable currently processed. This also makes the recipe more compact, since common keys, such as project class or time period, have to be defined only once and not repeated for all datasets. As in v1.0, the datasets

listed in the datasets section are processed for all diagnostics and variables defined in the recipes. Datasets to be used only by a specific diagnostic or providing only a specific variable can be added as additional datasets in the diagnostic or in the variable dictionary, respectively, using exactly the same syntax.

### 3.3 Preprocessors section

This is a new feature of ESMValTool v2.0: in the `preprocessors` section, one or more sets of preprocessing operations (preprocessors) can be defined. Each preprocessor is identified by a unique name and includes a list of operations and settings (see Sect. 4 for details). Once defined, a preprocessor can be applied to an arbitrary number of variables listed in the diagnostics section. This applies also when variable-specific settings are given in the preprocessor: it is possible, for example, to set a reference dataset as a target grid for the regridding operator with the reference dataset being different for each variable. When parsing the recipe, the tool automatically replaces these settings in the preprocessor definition with the corresponding variable settings, depending on the preprocessor-variable combination. The usage of the YAML format makes all these operations quite intuitive for the user and easy to implement for the developer. The preprocessors section in a recipe is optional and can be omitted if only the default preprocessing of the data is desired. The default preprocessor will apply fixes to the data (if required), perform CMOR compliance checks and select the data for the requested time frame only.

### 3.4 Diagnostics section

In the `diagnostics` section one or more diagnostics can be defined. Each diagnostic is identified by a name and contains one or more variables and one or more diagnostic scripts. The variables and the scripts are defined as subsections of the diagnostics section. This nested structure allows for the easy definition of complex diagnostics dealing with multiple variables and/or applying multiple diagnostics scripts to the same set of variables. Within the variable dictionary, additional settings can be defined, such as the preprocessor to be applied (as defined in the preprocessors section), the additional variable-specific datasets which are not included in the datasets section, and other variable-specific settings used by the diagnostic. The same can be done for the scripts dictionary by providing a list of settings to customize the run-time behavior of a diagnostic, together with the path to the diagnostic script itself. This feature replaces the language-specific `cfg` files that were used in ESMValTool v1.0 and allows the centralization of all user-configurable settings in a single file (the recipe). Note that the diagnostic scripts subsection can be left out, meaning that it is possible to only apply a given preprocessor to one or more variables without any further analysis, i.e. to use the ESMValTool just for preprocessing purposes.

### 3.5 Advanced recipe features

In an ESMValTool v2.0 recipe, it is also possible to make use of the anchor and reference capability of the YAML format in order to avoid code duplication by re-using already defined recipe blocks and to keep the recipe compact. A list of settings given in a diagnostics script dictionary can, for instance, be anchored and referenced in another script dictionary within the

same recipe, while changing only some of the settings in the list: a typical application is when the same diagnostic is applied to multiple variables using each time a different set of contour levels for each plot while keeping other settings identical.

Another feature is the possibility of defining ancestors, i.e. tasks that have to be completed before a given diagnostic can be run. This is useful for complex recipes in which a diagnostic collects and plots the results obtained by other diagnostics.
For example in `recipe_perfmetrics_CMIP5.yml`, the grading metrics for individual variables across many datasets are pre-calculated and then collected by another script which combines them into a portrait diagram.

## 4 Data preprocessing

A typical requirement for the analysis of output from ESMs is some preprocessing of the input data by a number of operators which are quite common to many analyses and include, for instance, temporal and spatial subsetting, vertical and horizontal
regridding, masking, multi-model statistics, etc. As mentioned in the introduction, in ESMValTool v1.0 these operations were performed in two different parts of the tool: at the preprocessor level (as part of the Python-based workflow manager controlling the ESMValTool) and at the diagnostic level (distributed across the various diagnostic scripts and only partly centralized in the ESMValTool language-specific libraries). In ESMValTool v2.0, the code for these preprocessing operations is moved from the diagnostic scripts to the completely rewritten and revised preprocessor within the ESMValCore package.
The structure of the revised preprocessor is schematically depicted in the light blue box in Fig. 1: each of the preprocessor functionalities is represented by a yellow box and can be controlled by a number of recipe settings, depicted by the purple tabs. Some operations require user-provided scripts, e.g. for variable derivation or fixes to the CMOR format, which are represented by the blue tabs. The figure shows the default order in which these operations are applied to the input data. This order has been defined in a way that minimizes the loss of information through the various steps, although it may not always be the optimal
choice in terms of performance (see also Sect. 4.6). For example, regridding and multi-model statistics are applied before temporal and spatial averaging. This default order can be changed and customized by the user in the ESMValTool recipe, although not all combinations are possible: multi-model statistics, for instance, can only be calculated after regridding the data.

The ESMValTool v2.0 preprocessor is entirely written in Python and takes advantage of the Iris library (v2.2.1) developed by the Met Office (Met Office, 2010-2019). Iris is an open-source, community-driven Python 3 package for analyzing and
visualizing Earth science data, building upon the rich software stack available in the modern scientific Python ecosystem. Iris supports reading several different popular scientific file formats, including netCDF, into an internal format based on the Climate and Forecast (CF) Metadata Convention (http://cfconventions.org/). Iris preserves the metadata that describe the data allowing users to handle their multi-dimensional data within a meaningful, domain-specific context and through a rich and expressive interface. Iris represents multi-dimensional data and the associated metadata for a single phenomenon through
the abstraction of a hypercube, also known as Iris cube, i.e. a multi-dimensional numerical array that stores the numerical values of a physical variable, coupled with a metadata object that fully describes the actual data. Iris cubes allow users to perform a powerful and extensive range of cube operations from simple unit conversion, subsetting and extraction, merging and concatenation to statistical aggregations and reductions, regridding and interpolation, and arithmetic operations. Internally,

Iris keeps pace with the modern developments provided by the scientific Python community, to ensure that users continue to benefit from advances in the Python ecosystem. In particular, Iris takes advantage of Dask (v2.3.0, https://dask.org/) to provide lazy evaluation (meaning that the actual data do not have to be loaded into the memory before they are really needed) and out-of-core processing, allowing Iris to perform at scale from efficient single-machine workflows through to multi-core clusters and HPC machines. One of the major advantages of Iris, which motivated its adoption for the revised ESMValTool preprocessor, is its ability to load large datasets as cubes and to pass these objects from one module to another and alter them as needed during the preprocessor workflow, while keeping all these stages in memory without need for time-intensive I/O operations. Each of the preprocessor modules is a Python function that takes an Iris cube and an optional set of arguments as input and returns a cube. The arguments controlling the operations to be performed by the modules are in most cases directly specified in the ESMValTool recipe. This makes it easy to read the recipe and also allows simple re-use of the ESMValTool preprocessor functions in other software.

In addition to Iris, NumPy (v1.17, https://numpy.org/) and SciPy (v1.3.1, https://www.scipy.org/) for generic array and mathematical/statistical operations, the ESMValTool preprocessor uses some specialized packages like Python-stratify (v0.1, https://github.com/SciTools-incubator/Python-stratify) for vertical interpolation, ESMPY (v7.1.0, https://www.earthsystemcog.org/projects/esmpy/) for regridding of irregular grids, and cf_units (v2.1.3, https://github.com/SciTools/cf-units) for standardization and conversion of physical units. Support for geographical maps is provided by the Cartopy library (v0.17.0, https://scitools.org.uk/cartopy/) and the Natural Earth dataset (v4.1.0, https://www.naturalearthdata.com/).

In the following, the ESMValTool preprocessor operations are described, together with their respective user settings. A detailed summary of these settings is given in Table 1.

## 4.1 Variable derivation

The variable derivation module allows calculation of variables which are not directly available in the input datasets. A typical example is total column ozone (`toz`) which is usually included in observational datasets (e.g., ESACCI-OZONE, Loyola et al., 2009; Lauer et al., 2017), but it is not part of the CMIP5 model data request. Instead the model output includes a three dimensional ozone concentration field. In this case, an algorithm to derive total column ozone from the ozone concentrations and air pressure is included in the preprocessor. Such algorithms can also be provided by the user. A corresponding custom CMOR table with the required variable metadata (standard_name, units, long_name, etc.) must also be provided, since this information is not available in the standard tables. Variable derivation is activated in the variable dictionary of the recipe setting the flag `derive: true`. Note that, by default, the preprocessor gives priority to existing variables in the input data before attempting to derive them, e.g. if a derived variable is already available in the observational dataset. This behavior can be changed by forcing variable derivation, i.e. the preprocessor will derive the variable even if it is already available in the input data, by setting `force_derivation: true`. The ESMValCore package currently includes derivation algorithms for 34 variables, listed in Table 2.

**Table 1.** Overview of the preprocessor functionalities and related recipe settings.

| Functionality | Key | Possible values | Description |
|---|---|---|---|
| Variable derivation (Sect. 4.1) | `derive` | `true`, `false` | Derive the variable from basic variables using a derivation function |
| | `force_derivation` | `true`, `false` | Force derivation even if the variable is available in the input data |
| CMOR check and fixes (Sect. 4.2) | | | Check CMOR compliance and apply dataset-specific fixes if required |
| Vertical interpolation (Sect. 4.3) | `levels` | (list of) level(s) [Pa] or [m] | Extract or interpolate at the given level(s) |
| | | dataset name | Extract or interpolate at the levels of the given dataset |
| | | `reference_dataset` | Extract or interpolate at the levels of the reference dataset |
| | | `alternative_dataset` | Extract or interpolate at the levels of the alternative dataset |
| | `scheme` | `linear` | Interpolate using a linear scheme |
| | | `nearest` | Interpolate using a nearest-neighbor scheme |
| | | `linear_extrapolate` | Interpolate using a linear scheme allowing for extrapolation |
| | | `nearest_extrapolate` | Interpolate using a nearest-neighbor scheme allowing for extrapolation |
| Land/Sea weighting (Sect. 4.4) | `area_type` | `land` | Weigh data by land fraction of the grid-cell |
| | | `sea` | Weigh data by sea fraction of the grid-cell |
| | `exclude` | (list of) dataset name(s) | Exclude the given dataset(s) from weighting |
| Land/Sea/Ice masking (Sect. 4.5) | `mask_out` | `land` | Set grid points with more than 50% land coverage to missing |
| | | `sea` | Set grid points with more than 50% sea coverage to missing |
| | | `ice` | Set grid points with more than 50% ice coverage to missing |
| | | `glaciated` | Set grid points with more than 50% of glaciers coverage to missing |
| Horizontal regridding (Sect. 4.6) | `target_grid` | NxM | Regrid to a $N° \times M°$ rectangular grid |
| | | dataset name | Regrid to the same grid of the given dataset |
| | | `reference_dataset` | Regrid to the same grid of the reference dataset |
| | | `alternative_dataset` | Regrid to the same grid of the alternative dataset |
| | `lat_offset` | `true`, `false` | Offset the grid centers of latitude by half a grid cell size |
| | `lon_offset` | `true`, `false` | Offset the grid centers of longitude by half a grid cell size |
| | `scheme` | `linear` | Regrid using linear regridding |
| | | `linear_extrapolate` | Regrid using linear regridding allowing for extrapolation |
| | | `nearest` | Regrid using nearest-neighbor regridding |
| | | `area_weighted` | Regrid using area-weighted regridding |
| | | `unstructured_nearest` | Regrid using nearest-neighbor regridding for unstructured data |
| Missing value masking (Sect. 4.7) | `threshold_frac` | [0,1] | Apply a uniform missing value mask |

| Functionality | Key | Possible values | Description |
|---|---|---|---|
| Detrend (Sect. 4.9) | dimension | dimension name | Detrend data along a given dimension. |
| | method | linear | Subtract the linear trend of the given dimension from the data |
| | | constant | Subtract the mean along the given dimension from the data |
| Multi-model statistics (Sect. 4.10) | statistics | mean | Calculate multi-model mean of the input datasets |
| | | median | Calculate multi-model median of the input datasets |
| | span | overlap | Consider only the overlapping time-period among all datasets |
| | | full | Consider the maximum time period covered by all datasets |
| | exclude | (list of) dataset name(s) | Exclude the given dataset(s) from the multi-model calculation |
| | | reference_dataset | Exclude the reference dataset from the multi-model calculation |
| | | alternative_dataset | Exclude the alternative dataset from the multi-model calculation |
| Temporal statistics (Sect. 4.8 and 4.11) | regrid_time | mon, day | Re-align time axis to new time units |
| | extract_time | | Extract time between start and end date |
| | start_year | any year | |
| | start_month | [1,12] | |
| | start_day | [1,31] | |
| | end_year | any year | |
| | end_month | [1,12] | |
| | end_day | [1,31] | |
| | extract_season | | Extract a specific season |
| | season | DJF, MAM, JJA, SON | |
| | extract_month | | Extract a specific month |
| | month | [1,12] | |
| | daily_statistics | | Apply daily statistics |
| | operator | mean,median,std_dev, min,max,sum | |
| | monthly_statistics | | Apply monthly statistics |
| | operator | mean,median,std_dev, min,max,sum | |
| | seasonal_statistics | | Apply seasonal statistics |
| | operator | mean,median,std_dev, min,max,sum | |

| Functionality | Key | Possible values | Description |
|---|---|---|---|
| | annual_statistics | | Apply annual statistics |
| | operator | mean,median,std_dev, min,max,sum | |
| | decadal_statistics | | Apply decadal statistics |
| | operator | mean,median,std_dev, min,max,sum | |
| | climate_statistics | | Apply climate statistics |
| | operator | mean,median,std_dev, min,max,sum | |
| | period | full,season,month,day | Calculate full, seasonal, monthly or daily climatology |
| | anomalies | | Calculate anomalies |
| | period | full,season,month,day | Calculate full, seasonal, monthly or daily anomaly |
| Spatial statistics | extract_region | | Extract a rectangular region given the limits |
| (Sect. 4.8 and 4.11) | start_latitude | [-90, 90] | |
| | start_longitude | [0, 360] | |
| | end_latitude | [-90, 90] | |
| | end_longitude | [0, 360] | |
| | extract_named_regions | | Extract a predefined named region |
| | regions | a (list of) named region(s) | |
| | extract_shape | | Extract one or more shapes or a representative point for these shapes |
| | shapefile | path to shape file | |
| | method | contains | Select all points contained by the shape |
| | | representative | Select a single representative point of the shape |
| | crop | true, false | Crop (true) or mask (false) the selected shape |
| | decomposed | true, false | Mask the regions in the shape files separately, adding an extra dimension |
| | extract_volume | | Extract a depth range |
| | z_min | depth [m] | |
| | z_max | depth [m] | |
| | extract_transect | | Extract a transect at a given latitude or longitude |
| | latitude | [-90,90] | |
| | longitude | [0,360] | |

| Functionality | Key | Possible values | Description |
|---|---|---|---|
| | `extract_trajectory` | | Extract a transect along the given trajectory |
| | `latitude_points` | list of latitudes | |
| | `longitude_points` | list of longitudes | |
| | `number_point` | n. of points to interpolate | |
| | `zonal_statistics` | | Apply statistics along the longitude axis |
| | `operator` | `mean,median,std_dev,` `min,max,sum` | |
| | `meridional_statistics` | | Apply statistics along the latitude axis |
| | `operator` | `mean,median,std_dev,` `min,max,sum` | |
| | `area_statistics` | | Apply statistics along the latitude and longitude axes |
| | `operator` | `mean,median,std_dev,` `min,max,sum` | |
| | `volume_statistics` | | Calculate the volume-weighted average of a 3D field |
| | `operator` | `mean` | |
| | `depth_integration` | | Calculate the volume-weighted z-dimensional sum |
| Unit conversion (Sect. 4.12) | `units` | a UDUNITS [a] string | Convert units of the input data |

[a] https://www.unidata.ucar.edu/software/udunits/

## 4.2 CMOR check and fixes

Similar to ESMValTool v1.0, the CMOR check module checks for compliance of the netCDF input data with the CF metadata convention and CMOR standards used by the ESMValTool. As in v1.0, it checks for the most common dataset problems (e.g., coordinate names and ordering, units, missing values, etc.) and includes a number of project-, dataset- and variable-specific
fixes to correct these known errors. In v1.0, the format checks and fixes were based on the CMOR tables of the CMIP5 project (https://github.com/PCMDI/cmip5-cmor-tables). This has now been extended and allows the use of CMOR tables from different projects (like CMIP5, CMIP6, obs4mips, etc.) or user-defined custom tables (required in case of derived variables which are not part of an official data request, see Sect. 4.1). The CMOR tables for the supported projects are distributed together with the ESMValCore package, using the most recent version available at the time of the release. The adoption of Iris
with strict requirements of CF compliance for input data, required the implementation of fixes for a larger number of datasets compared to v1.0. Although from a user's perspective this makes the reading of some datasets more demanding, the stricter standards enforced in the new version of the tool ensure their correct interpretation and reduce the probability of unintended behavior or errors.

## 4.3 Level selection and vertical interpolation

Reducing the dimensionality of input data is a common task required by diagnostics. Three-dimensional fields are often analyzed by first extracting two-dimensional data at a given level. In the preprocessor, level selection can be performed on any input data containing a vertical coordinate, like pressure level, altitude or depth. One or more levels can be specified by the `levels` key in the preprocessors section of the recipe: this may be a (list of) numerical value(s), a dataset name whose vertical coordinate can be used as target levels for the selection, or a predefined set of CMOR standard levels. If the requested level(s)
is (are) not available in the input data, a vertical interpolation will be performed among the available input levels. In this case, the interpolation scheme (linear or nearest neighbor) can be specified as a recipe setting (`scheme`), and extrapolation can be enabled or disabled. The interpolation is performed by the Python-stratify package which, in turn, uses a C library for optimal computational performance. This operation preserves units and masking patterns.

## 4.4 Land/Sea fraction weighting

Several land surface variables, for example fluxes for the carbon cycles, are reported as mass per unit area, where the area refers to land surface area and not grid-box area. When globally or regionally integrating these variables, weighting by both the surface quantity and the land/sea fraction has to be applied. The preprocessor implements such weighting, by multiplying the given input field by a fraction in the range 0-1, to account for the fact that not all grid points are completely land- or sea-covered. This preprocessor allows to specify whether `land` or `sea` fraction weighting has to be applied and also gives the
possibility to exclude those datasets for which no information about the land/sea fraction is available.

## 4.5 Land/Sea/Ice masking

The masking module allows to extract specific data domains, such as land-, sea-, ice- or glacier-covered regions, as specified by the `mask_out` setting in the recipe. The grid points in the input data corresponding to the specified domain are masked out by setting their value to missing, i.e. using the netCDF attribute `_FillValue`. The masking module uses the CMOR fx-variables to extract the domains. These variables are usually part of the data requests of CMIP and other projects, and therefore have the advantage of being on the same grid as their corresponding models. For example, the fx-variables `sftlt` and `sftot` are used to define land- or sea-covered regions, respectively, on regular and irregular grids. In case these variables are not available for a given dataset, as is often the case for observational datasets, the masking module uses the Natural Earth shape files to generate a mask at the corresponding horizontal resolution. This latter option is currently only available for regular grids.

## 4.6 Horizontal regridding

Working with a common horizontal grid across a collection of datasets is a very important aspect of multi-model diagnostics and metric computations. Although model and observational datasets are provided at different native grid resolutions, it is often required to scale them to a common grid in order to apply diagnostic analyses, as the root-mean-square error (RMSE) at each grid point, or to calculate multi-model statistics (see Sect. 4.10). This operation is required both from a numerical point of view (common operators can not be applied on numerical data arrays with different shapes) and from a statistical point of view (different grid resolutions imply different Euclidian norms, hence data from each model has different statistical weights). The regridding module can perform horizontal regridding onto user-specified target grids (`target_grid`) with a number of interpolation schemes (`scheme`) available. The target grid can either be a standard regular grid with a resolution of $M \times N$ degrees, or the grid of a given dataset (for example, the reference dataset). Regridding is then performed via interpolation.

While the target grid is often a standard regular grid, the source grids exhibit a larger variety. Particularly challenging are grids where the native grid coordinates do not coincide with standard latitudes and longitudes, often referred to as irregular grids, although varying terminology exists. As a consequence, the relationship between source and target grid cells can be very complex. Such irregular grids are common for ocean data, where the poles are placed over land to avoid the singularities in the computational domain, thereby distorting the resulting grid. Irregular grids are also commonly used for map projections of regional models. As long as these grids exhibit a rectangular topology, data living on them can still be stored in cubes and the resulting coordinates in the latitude-longitude coordinate system can be provided in standardized form as auxiliary coordinates following the CF conventions. For CMIP data, this is mandatory for all irregular grids. The regridding module uses this information to perform regridding between such grids, allowing, for example, for the easy inclusion of ocean data in multi-model analyses.

The regridding procedure also accounts for masked data, meaning that the same algorithms are applied while preserving the shape of the masked domains. This can lead to small numerical errors, depending on the domain under consideration and its shape. The choice of the correct regridding scheme may be critical in case of masked data. Using an inappropriate option may alter the mask significantly and thus introduce a large bias in the results. For example, bilinear regridding uses the nearest grid

points in both horizontal directions to interpolate new values. If one or more of these points are missing, calculation is not possible and a missing value is assigned to the target grid cell. This procedure always increases the size of the mask, which can be particularly problematic for areas where the original mask is narrow, e.g. islands or small peninsulas in case of land/sea masking. A much more recommended scheme in this case is nearest-neighbor regridding. This option approximately preserves the mask, resulting in smaller biases compared to the original grid. Depending on the target grid, the area-weighted scheme may also be a good choice in some cases. The most suitable scheme is strongly dependent on the specific problem and there is no one-fits-all solution. The user needs to be aware that regridding is not a trivial operation which may lead to systematic errors in the results. The available regridding schemes are listed in Table 1.

## 4.7 Missing value masking

When comparing model data to observations, the different data coverage can introduce significant biases (e.g., de Mora et al., 2013). Coverage of observational data is often incomplete. In this case, the calculation of even simple metrics like spatial averages could be biased, since a different number of grid-boxes are used for the calculations if data are not consistently masked. The preprocessor implements a missing values masking functionality, based on an approach which was originally part of the "performance metrics" routines of ESMValTool v1.0. This approach has been implemented in Python as a preprocessor function, which is now available to all diagnostics. The missing value masking requires the input data to be on the same horizontal and vertical grid, and therefore must necessarily be performed after level selection and horizontal regridding. The data can, however, have different temporal coverage. For each grid point, the algorithm considers all values along the time coordinate (independently of its size) and the fraction of such values which are missing. If this fraction is above a user-specified threshold (`threshold_fraction`) the grid point is left unchanged, otherwise it is set to missing along the whole time coordinate. This ensures that the resulting masks are constant in time and allows masking datasets with different time coverage. Once the procedure has been applied to all input datasets, the resulting time-independent masks are merged to create a single mask which is then used to generate consistent data coverage for all input datasets. In case of multiple selected vertical levels, the missing values masks are assembled and applied to the grid independently at each level.

This approach minimizes the loss of generality by applying the same threshold to all datasets. The choice of the threshold strongly depends on the datasets used and on their missing value patterns. As a rule of thumb, the higher the number of missing values in the input data, the lower the threshold, which means that the selection along the time coordinate must be less strict in order to preserve the original pattern of valid values and to avoid completely masking out the whole input field.

## 4.8 Temporal and spatial subsetting

All basic time extraction and concatenation functionalities have been ported from v1.0 to the v2.0 preprocessor and have not changed significantly. Their purpose is to retrieve the input data and extract the requested time range as specified by the keys `start_year` and `end_year` for each of the dataset dictionaries of the ESMValTool recipe (see Sect. 3 for more details). If the requested time range is spread over multiple files, a common case in the CMIP5 data pool, the preprocessor concatenates the data before extracting the requested time period. An important new feature of time concatenation is the possibility to

concatenate data across different model experiments. This is useful, for instance, to create time-series combining the CMIP historical experiment with a scenario projection. This option can be set by defining the `exp` key of the dataset dictionary in the recipe as a Python list, e.g. `[historical, rcp45]`. These operations are only applied while reading the original input data.

More specific functions are applied during the preprocessing phase to extract a specific subset of data from the full dataset. This extraction can be done along the time axis, in the horizontal direction or in the vertical direction. These functions generally reduce the dimensionality of data. Several extraction operators are available to subset the data in time (`extract_time`, `extract_season`, `extract_month`) and in space (`extract_region`, `extract_named_regions`, `extract_shape`, `extract_volume`, `extract_transect`, `extract_trajectory`), see again Table 1 for details.

### 4.9 Detrend

Detrending is a very common operation in the analysis of time series. In the preprocessor, this can be applied along any dimension in the input data, although the most usual case is detrending along the time axis. The `method` used for detrending can be either `linear` (the linear trend along the given dimension is calculated and subtracted from the data) or `constant` (the mean along the given dimension is calculated and subtracted from the data).

### 4.10 Multi-model statistics

Computing multi-model statistics is an integral part of model analysis and evaluation: individual models display a variety of biases depending, for instance, on model configurations, initial conditions, forcings and implementation. When comparing model data to observational data, these biases are typically smaller when multi-model statistics are considered. The preprocessor has the capability of computing a number of multi-model statistical measures: using the `multi_model_statistics` module enables the user to calculate either a multi-model mean, median or both, that are passed as additional dataset(s) to the diagnostics. Additional statistical operators (e.g., standard deviation) can be easily added to the module if required. Multi-model statistics are computed along the time axis and, as such, can be computed across a common overlap in time or across the full length in time of each model: this is controlled by the `span` argument. Note that in case the full-length case is used, the number of datasets actually used to calculate the statistics can vary along the time coordinate if the datasets cover different time ranges. The preprocessor function is capable of excluding any dataset in the multi-model calculations (option `exclude`): a typical example is the exclusion of the observational dataset from the multi-model calculations. Model datasets must have consistent shapes, which is needed from a statistical point of view since weighting is not yet implemented. Furthermore, data with a dimensionality higher than four (time, vertical axis, two horizontal axes) is also not supported.

### 4.11 Temporal and spatial statistics

Changing the spatial and temporal dimensions of model and observational data is a crucial part of most analyses. In addition to the subsetting described in Sect. 4.8, a second general class of preprocessor functions applies statistical operators along a temporal (`daily_statistics`, `monthly_statistics`, `seasonal_statistics`, `annual_statistics`, `decadal_statistics`,

**Table 2.** List of variables for which derivation algorithms are available in ESMValTool v2.0, and the corresponding input variables. ISCCP is the International Satellite Cloud Climatology Project, TOA means top-of-the-atmosphere.

| Derived variable | Description | Realm | Input variables for derivation |
|---|---|---|---|
| alb | Albedo at the surface | atmosphere | rsds, rsus |
| amoc | Atlantic Meridional Overturning Circulation | ocean | msftmyz |
| asr | Absorbed shortwave radiation | atmosphere | rsdt, rsut |
| clhmtisccp | ISCCP high level medium-thickness cloud area fraction | atmosphere | clisccp |
| clhtkisccp | ISCCP high level thick cloud area fraction | atmosphere | clisccp |
| cllmtisccp | ISCCP low level medium-thickness cloud area fraction | atmosphere | clisccp |
| clltkisccp | ISCCP low level thick cloud area fraction | atmosphere | clisccp |
| clmmtisccp | ISCCP middle level medium-thickness cloud area fraction | atmosphere | clisccp |
| clmtkisccp | ISCCP middle level thick cloud area fraction | atmosphere | clisccp |
| ctotal | Total carbon mass in ecosystem | land | cVeg, cSoil |
| et | Evapotranspiration | atmosphere | hfls |
| lvp | Latent heat release from precipitation | atmosphere | hfls, ps, evspsbl |
| lwcre | TOA longwave cloud radiative effect | atmosphere | rlut, rlutcs |
| lwp | Liquid water path | atmosphere | clwvi, cliwi |
| netcre | TOA net cloud radiative effect | atmosphere | rlut, rlutcs, rsut, rsutcs |
| ohc | Heat content in grid cell | ocean | thetao, volcello |
| rlns | Surface net downward longwave radiation | atmosphere | rlds, rlus |
| rlnst | Net atmospheric longwave cooling | atmosphere | rlds, rlus, rlut |
| rlnstcs | Net atmospheric longwave cooling assuming clear sky | atmosphere | rldscs, rlus, rlutcs |
| rlntcs | TOA net downward longwave radiation assuming clear sky | atmosphere | rlutcs |
| rsns | Surface net downward shortwave radiation | atmosphere | rsds, rsus |
| rsnst | Heating from shortwave absorption | atmosphere | rsds, rsdt, rsus, rsut |
| rsnstcs | Heating from shortwave absorption assuming clear sky | atmosphere | rsdscs, rsdt, rsuscs, rsutcs |
| rsnstcsnorm | Heating from shortwave absorption assuming clear sky normalized by incoming solar radiation | atmosphere | rsdscs, rsdt, rsuscs, rsutcs |
| rsnt | TOA net downward shortwave radiation | atmosphere | rsdt, rsut |
| rsntcs | TOA net downward shortwave radiation assuming clear sky | atmosphere | rsdt, rsutcs |
| rtnt | TOA net downward total radiation | atmosphere | rsds, rsut, rlut |
| sispeed | Speed of ice to account for back-and-forth movement of the ice | sea-ice | siu, siv |
| sithick | Sea ice thickness | sea-ice | sit, sic |
| sm | Volumetric moisture in upper portion of soil column | land | mrsos |
| swcre | TOA shortwave cloud radiative effect | atmosphere | rlut, rlutcs, rsut, rsutcs |
| toz | Total ozone column | atmosphere | tro3, ps |
| uajet | Jet position expressed as latitude of maximum meridional wind speed | atmosphere | ua |
| vegfrac | Vegetation fraction | land | baresoilFrac |

climate_statistics, anomalies) or spatial (zonal_statistics, meridional_statistics, area_statistics, volume_statistics) axis. The statistical operators allow the calculation of mean, median, standard deviation, minimum, and maximum along the given axis (with the exception of volume_statistics which only supports the mean). An additional operator, depth_integration, calculates the volume-weighted $z$-dimensional sum of the input cube. Like the subsetting operators (Sect. 4.8), these also sig-
nificantly reduce the size of the input data passed to the diagnostics for further analysis and plotting.

## 4.12 Unit conversion

In ESMValTool v2.0, input units always follow the CMOR definition, which is not always the most convenient for plotting. Degree Celsius, for instance, is for some analyses more convenient that the standard Kelvin unit. Using the cf_units Python package, the unit conversion module of the preprocessor can convert the physical unit of the input data to a different one, as
given by the units argument. This functionality can also be used to make sure that units are identical across all datasets before applying a diagnostic.

## 5   Additional features

## 5.1   CMORization of observational datasets

As discussed in Sect. 4.2, the ESMValTool requires the input data to be in netCDF format and to comply with the CF meta-
data convention and CMOR standards. Observational and reanalysis products in the standard CF/CMOR format are available via the obs4mips (https://esgf-node.llnl.gov/projects/obs4mips/) and ana4mips (https://esgf.nccs.nasa.gov/projects/ana4mips/) projects, respectively (see also Teixeira et al., 2014). Their use is strongly recommended, when possible. Other datasets not available in these archives can be obtained by the user from the respective sources and reformatted to the CF/CMOR standard using the cmorizers included in the ESMValTool. The cmorizers are dataset-specific scripts that can be run once to generate
a local pool of observational datasets for usage with the ESMValTool, since no observational datasets are distributed with the tool. Supported languages for cmorizers are Python and NCL. These scripts also include detailed instructions on where and how to download the original data and serve as templates to create new cmorizers for datasets not yet included. The current version features cmorizing scripts for 46 observational and reanalysis datasets. As in v1.0, the observational datasets are grouped in Tiers, depending on their availability: Tier 1 (for obs4mips and ana4mips datasets), Tier 2 (for other freely-available datasets),
and Tier 3 (for restricted datasets, i.e. dataset which requires a registration to be downloaded or that can only be obtained upon request by the respective authors). An overview of the Tier 2 and Tier 3 datasets for which a cmorizing script is available in ESMValTool v2.0 is given in Table 3. Note that observational datasets cmorized for ESMValTool v1.0 may not be directly working with v2.0, due to the much stronger constraints on metadata set by the Iris library.

**Table 3.** List of the observational and reanalysis datasets for which a cmorizing script is available in ESMValTool *v2.0*, together with the corresponding variables (realms), Tier level and reference.

| Dataset | Variable (realm) | Tier | Reference |
|---|---|---|---|
| AURA-TES | `tro3` (atmosphere) | 3 | Beer (2006) |
| CDS-SATELLITE-LAI-FAPAR | `fapar, lai` (land) | 3 | Baret et al. (2007) |
| CDS-SATELLITE-SOIL-MOISTURE | `sm, smStderr` (land) | 3 | Gruber et al. (2019) |
| CDS-UERRA | `sm` (land) | 3 | Ridal et al. (2017) |
| CDS-XCH4 | `xch4` (atmosphere) | 3 | Buchwitz et al. (2018) |
| CDS-XCO2 | `xco2` (atmosphere) | 3 | Buchwitz et al. (2018) |
| CERES-EBAF | `rlut, rlutcs, rsut, rsutcs` (atmosphere) | 2 | Loeb et al. (2018) |
| CERES-SYN1deg | `rlds, rldscs, rlus, rluscs, rlut, rlutcs, rsds, rsdscs, rsdt, rsus, rsuscs, rsut, rsutcs` (atmosphere) | 3 | Wielicki et al. (1996) |
| CRU | `pr, tas` (atmosphere) | 2 | Harris et al. (2014) |
| Duveiller2018 | `albDiffiTr13` | 2 | Duveiller et al. (2018) |
| Eppley-VGPM-MODIS | `intpp` (ocean) | 2 | Behrenfeld and Falkowski (1997) |
| ERA5 | `clt, pr, prsn, ps, psl, ptype, rls, rlds, rsds, rsdt, rss, uas, vas, tas, tasmax, tasmin, tdps, ts, tsn` (atmosphere), `evspsbl, evspsblpot, mrro` (land), `orog` (fx) | 3 | C3S (2017) |
| ERA-Interim | `clivi, clt, clwvi, hfds, hur, hus, pr, prsn, prw, ps, psl, rlds, rsds, rsdt, rss, ta, tas, tasmax, tasmin, tauu, tauv, tdps, ts, ua, uas, va, vas, wap, zg` (atmosphere), `evspsbl, tsn` (land), `orog, sftlf` (fx), `hfds, tos` (ocean) | 3 | Dee et al. (2011) |
| ERA-Interim-Land | `sm` (Lmon) | 3 | Balsamo et al. (2015) |
| ESACCI-AEROSOL | `abs550aer, od550aer, od550aerStderr, od550lt1aer, od870aer, od870aerStderr` (aero) | 2 | Popp et al. (2016) |
| ESACCI-CLOUD | `clivi, clt, cltStderr, clwvi` (atmosphere) | 2 | Stengel et al. (2017) |
| ESACCI-FIRE | `burntArea` (land) | 2 | Chuvieco et al. (2016) |
| ESACCI-LANDCOVER | `baresoilFrac, cropFrac, grassFrac, shrubFrac, treeFrac` (land) | 2 | Defourny (2016) |
| ESACCI-OC | `chl` (ocean) | 2 | Sathyendranath et al. (2016) |
| ESACCI-OZONE | `toz, tozStderr, tro3prof, tro3profStderr` (atmosphere) | 2 | Loyola et al. (2009) |
| ESACCI-SOILMOISTURE | `dos, dosStderr, sm, smStderr` (land) | 2 | Liu et al. (2011, 2012) |

| Dataset | Variable (realm) | Tier | Reference |
|---|---|---|---|
| ESACCI-SST | ts, tsStderr (atmosphere) | 2 | Merchant et al. (2014) |
| FLUXCOM | gpp (land) | 3 | Jung et al. (2019) |
| GCP | nbp (land), fgco2 (ocean) | 3 | Quéré et al. (2018) |
| GHCN | pr (atmosphere) | 2 | Jones and Moberg (2003) |
| HadCRUT3 | tas, tasa (atmosphere) | 2 | Brohan et al. (2006) |
| HadCRUT4 | tas, tasa (atmosphere) | 2 | Morice et al. (2012) |
| HadISST | tos, sic (ocean), ts (atmosphere) | 2 | Rayner et al. (2003) |
| HWSD | cSoil (land), areacella, sftlf (fx) | 2 | Wieder (2014) |
| ISCCP-FH | alb, prw, ps, rlds, rlus, rlut, rlutcs, rsds, rsdt, rsus, rsut, rsutcs, tas, ts (atmosphere) | 2 | Zhang et al. (2019) |
| JMA-TRANSCOM | nbp (land), fgco2 (ocean) | 3 | Maki et al. (2010) |
| LAI3g | lai (land) | 3 | Zhu et al. (2013) |
| LandFlux-EVAL | et, etStderr (land) | 3 | Mueller et al. (2013) |
| Landschuetzer2016 | fgco2, spco2, dpco2 (ocean) | 2 | Landschützer et al. (2016) |
| MERRA2 | sm (land) | 3 | Gelaro et al. (2017) |
| MODIS | cliwi, clt, clwvi, iwpStderr, lwpStderr (atmosphere), od550aer (aero) | 3 | Platnick et al. (2003); Levy et al. (2013) |
| MTE | gpp, gppStderr (land) | 3 | Jung et al. (2011) |
| NCEP | hur, hus, pr, rlut, ta, tas, ua, va, wap, zg (atmosphere) | 2 | Kalnay et al. (1996) |
| NDP | cVeg (land) | 3 | Gibbs (2006) |
| NIWA-BS | toz, tozStderr (atmosphere) | 3 | Bodeker et al. (2005) |
| NSIDC-0116 | usi, vsi (sea-ice) | 3 | Tschudi (2019) |
| PATMOS-x | clt (atmosphere) | 2 | Heidinger et al. (2014) |
| PHC | thetao, so (ocean) | 2 | Steele et al. (2001) |
| PIOMAS | sit (ocean) | 2 | Zhang and Rothrock (2003) |
| UWisc | clwvi, lwpStderr (atmosphere) | 3 | O'Dell et al. (2008) |
| WOA | no3, o2, po4, si, so, thetao (ocean) | 2 | Locarnini et al. (2013) |

## 5.2 Provenance and tags

ESMValTool v2.0 contains a large number of recipes that perform a wide range of analyses on many different scientific themes (see the companion papers). Depending on the application, sorting and tracking of the scientific output (plots and netCDF files) produced by the tool can therefore be quite challenging. To simplify this task, ESMValTool v2.0 implements a provenance and tagging system that allows to document and organize the results, while keeping track of all the input data used to produce them (reproducibility and transparency of the results).

Provenance information is generated using the W3C-PROV reference format and collected at run-time. It is then attached to any output (plots and netCDF files) produced by the tool and is also saved to a separate log file. Using the W3C-PROV format ensures that the ESMValTool provenance is compatible with other (external) tools for viewing and processing provenance information. Examples of stored information include: all global attributes of input netCDF files, preprocessor settings, diagnostic script settings, and software version numbers. Along with this rather technical information, a set of scientific provenance tags are available. These include, for example, diagnostic script name and recipe authors, funding projects, references for citation purposes, as well as tags for categorizing the result plots into various scientific topics (like chemistry, dynamics, sea-ice, etc.) realms (land, atmosphere, ocean, etc.) or statistics applied (RMSE, anomaly, trend, climatology, etc.). This facilitates the publication and browsing of the ESMValTool output on web-pages, like the ESMValTool-based CMIP6 results browser hosted by the ESGF node at the Deutsches Klima RechenZentrum (DKRZ, https://cmip-esmvaltool.dkrz.de/), where model developers and users can inspect the results and filter them according to their scientific interests.

## 5.3 Automated testing and coding standards

To ensure code stability, maintainability and quality, the ESMValCore package and the installation procedures are automatically tested on a continuous integration server (CircleCI, https://circleci.com/) every time a change to the source code is pushed to the GitHub repository, making sure that these core components are reliable. Furthermore, static code analysis is performed by Codacy (https://www.codacy.com/) on all Python code, to identify possible sources of error without requiring any extra effort by the developers. Less strict static code analysis and basic requirements for the code formatting style is implemented for the diagnostics of the ESMValTool in form of a unit test, to enforce a clean, uniform-looking and easy-to-read code for all supported languages (Python, NCL, R and Julia). Code reviewers are encouraged to make use of the CircleCI and Codacy results to ensure that all contributions to the ESMValTool are reliable and can be maintained in the future with reasonable effort. CircleCI and Codacy offer free services for open source projects. We use these services to run open source software that could equally easily be run on other infrastructure. On CircleCI the unit tests are run in a Debian Linux docker container with a minimal version of Anaconda pre-installed (https://hub.docker.com/r/continuumio/miniconda3). On Codacy we make use of the various open source Python linters that are bundled into Prospector (https://prospector.readthedocs.io). These tools can also be installed and used on contributors' own computers with a minimal effort, as described in our contribution guidelines.

**Table 4.** Times required for running `recipe_perfmetrics_CMIP5.yml` with ESMValTool v1.1.0 and v2.0 using different numbers of maximum parallel tasks. Note that v1.0 did not support parallelization. The corresponding maximum memory usage as diagnosed in v2.0 is also shown. Each number in this table corresponds to the median of 10 ESMValTool runs, to account for the variability in the performance across different nodes. The nodes used for this analysis feature 24 physical cores.

| Number of parallel tasks | Run-time v1.0 [mins] | Run-time v2.0 [mins] | Max. memory usage v2.0 [Gb] |
| --- | --- | --- | --- |
| 1 (serial) | 534.1 | 177.1 | 41.5 |
| 2 | – | 78.7 | 41.8 |
| 4 | – | 45.2 | 44.1 |
| 8 | – | 27.4 | 54.0 |
| 16 | – | 19.6 | 62.4 |
| 32 | – | 16.6 | 66.9 |
| 64 | – | 16.5 | 74.7 |
| 68 (max) | – | 16.2 | 75.0 |

## 6  Performance and scaling tests

To demonstrate the improved performance of ESMValTool v2.0 over its predecessor version, a benchmark test has been performed for a representative recipe. The test was performed on the post-processing nodes of the Mistral Supercomputer at the DKRZ (see https://www.dkrz.de/up/systems/mistral for more details).

The ESMValTool `recipe_perfmetrics_CMIP5.yml` (see Supplement) is used as a benchmark and compared with the corresponding namelist of v1.1.0 (Eyring et al., 2016c). This recipe (namelist) is used as a test case as it represents all typical operations performed by the ESMValTool fairly well. For consistency, the recipe (namelist) in the two ESMValTool versions being compared contains exactly the same diagnostics and variables and is applied to the same datasets (models and observations) over identical time periods. The results produced with this setup are identical in v1.1.0 and v2.0. Since ESMValTool v1.1.0 did

not support parallel execution, the performances of the two versions in running this recipe (namelist) can be only compared in serial mode. For v2.0, benchmarking results are further analyzed using an increasing number of parallel tasks to demonstrate the gain in run-time when taking advantage of this new feature.

The benchmarking results are summarized in Table 4 and show that already in serial mode the time required to run the recipe with the new version is reduced by about a factor of three. Taking advantage of the task-based parallelization capability of v2.0,

the performance can be further improved. This allows reducing the run-time up to a maximum of a factor of about 33 with respect to v1.1.0 when using parallel capabilities. The maximum theoretical performance is obtained when all recipe tasks (68 in this example) are executed in parallel. Note however, that the run-time is limited by the slowest task in the recipe, which acts as a bottleneck. As shown in Table 4, this implies that no significant gain is obtained for this recipe when increasing the number of parallel tasks above 32. A further aspect that needs to be considered here is that increasing the number of parallel

tasks requires a larger amount of memory (last column in Table 4), since data from all tasks running simultaneously must be

stored in memory at the same time. The optimal choice of the number of parallel tasks to be used depends, therefore, on the total number of tasks in the recipe, on the differences in their individual run-times, and on the amount of memory available on the machine in use. Memory-intensive recipes, for instance, may require to be executed with a small number of parallel tasks on machines with limited memory, at the expense of the recipe run-time.

Since the task manager of v2.0 prioritizes tasks which are listed first in the recipe, the user can optimize the execution times by placing the more time-consuming tasks at the beginning of the recipe, especially when the execution times of individual tasks vary greatly and when ESMValTool is run with a number of parallel tasks which is significantly smaller than the total number of tasks performed by the recipe.

## 7   Summary

A new version of the ESMValTool has been developed to address the challenges posed by the increasing data volume of simulations produced by Earth System Models as contributions to large model intercomparison projects, such as CMIP6. The code of ESMValTool v2.0 has been completely restructured and now includes an independent Python package (ESMValCore), which features core functionalities such as the task manager, a revised preprocessor and an improved interface. The set of diagnostic scripts implementing scientific analysis on a wide range of Earth System Model variables and realms has also been

extended and is described in the companion papers Eyring et al. (2019), Lauer et al. (2019) and Weigel et al. (2019).

The redesigned ESMValCore package and its implementation in ESMValTool v2.0 resulted in significant improvements for both users (improved user-friendliness and more customization options) and developers (better code readability and easier maintenance). Benchmark tests performed with a representative ESMValTool recipe demonstrated the huge improvement in terms of performance (run-time) achieved by this new version: in serial mode it is already a factor of 3 faster than the previous

ESMValTool version and can be even faster when executed in parallel, with a factor of more than 30 reduction in run-time attainable on powerful compute resources. The centralization of the preprocessing operations in a core package also facilitated further optimization of the code, the possibility of running the ESMValTool in parallel and higher consistency among different diagnostics (e.g. regridding and masking of data). The revised and simplified interface also enables an easy installation and configuration of the ESMValTool for running at high-performance computing centers where data are stored, such as the super

nodes of the ESGF. In addition to the technical improvements discussed in this paper, ESMValTool v2.0 also features many new diagnostics and metrics which are discussed in detail in the three companion papers: Eyring et al. (2019), Lauer et al. (2019) and Weigel et al. (2019).

The ESMValTool undergoes continuous development, and additional improvements are constantly being implemented or planned for future releases. These include (but are not limited to):

– an increased flexibility of the CMOR check module of the preprocessor, allowing for the automatic recognition and correction of more errors in the input datasets, thus making reading of data more flexible, especially for data which are not part of any CMIP data request;

- more regridding options featuring, for example, masking options beyond the standard CMOR-fx masks of the CMIP data request, especially for irregular grids;

- a new preprocessor module for model ensemble statistics, reducing the amount of input data to be processed in multi-ensemble analyses;

- the increased usage of Dask arrays in preprocessor functions to keep the memory requirements low and further improve the performance;

- the possibility of reusing the output produced by specific preprocessor-variable combinations across different diagnostics, thus further improving the ESMValTool performance, while also reducing the disk space requirements;

- linking external tools such as the Community Intercomparison Suite (Watson-Parris et al., 2016, CIS) to the ESMValTool
  to target more specific topics, such as the spatial and temporal co-location of model and satellite data.

Note that some of the above issues could in principle already be addressed in ESMValTool v2.0 at the diagnostic level, but being general purpose functionalities, their implementation should take place in ESMValCore, where high quality code standard and testing will ensure their correct implementation.

The ESMValTool is a community development with currently more than 100 developers that contribute to the code. The
wider climate community is encouraged to use the ESMValTool and to participate in this development effort, by joining the ESMValTool development team for contributions of additional more in-depth diagnostics for evaluation of Earth System Models.

## 8   Code availability

ESMValTool (v2.0) is released under the Apache License, VERSION 2.0. The latest release of ESMValTool v2.0 is pub-
licly available on Zenodo at https://doi.org/10.5281/zenodo.3401363. The source code of the ESMValCore package, which is installed as a dependency of the ESMValTool v2.0, is also publicly available on Zenodo at https://doi.org/10.5281/zenodo. 3387139. ESMValTool and ESMValCore are developed on the GitHub repositories available at https://github.com/ESMValGroup.

*Author contributions.* MR designed the concept of the new version, coordinated its development and testing, contributed some parts of the code and wrote the paper. BA wrote most of the code, contributed to the concept of the new version, and reviewed code contributions by other developers. VE designed the concept. AL designed the concept, contributed some parts of the code and its testing. VP and JVR wrote the code for most of the preprocessor modules. LB and BB contributed some parts of the code and its testing. LdM wrote the code for

the preprocessor modules for temporal and spatial operations and contributed to documentation. FD developed and tested the installation procedure. LD contributed to the concept and active support with the Iris package. ND contributed to the technical architecture of the new version, to the initial prototype for the new preprocessor, and to the testing. PE contributed to the concept. BH contributed to documenting and testing the code. NK contribute to the concept and to testing. BL contributed to the concept and active support with the Iris package. SLT contributed to the preprocessor module for CMOR check. MS contributed some parts of the code, to its testing and documentation. KZ

wrote the code for the two preprocessor modules for irregular regridding and unit conversion. All authors contributed to the text.

*Competing interests.* The authors declare that they have no conflict of interest.

*Acknowledgements.* This work was funded by the Copernicus Climate Change Service (C3S) "Metrics and Access to Global Indices for Climate Projections (C3S-MAGIC)" project. ECMWF implements this Service on behalf of the European Commission. Funding was also provided by the EU Horizon 2020 research and innovation programme under the grant agreement No 824084 (IS-ENES3 project); by the

EU Horizon 2020 project "Coordinated Research in Earth Systems and Climate: Experiments, kNowledge, Dissemination and Outreach (CRESCENDO)"; and by Projects S1 (Diagnosis and Metrics in Climate Models) of the Collaborative Research Centre TRR 181 "Energy Transfer in Atmosphere and Ocean" funded by the Deutsche Forschungsgemeinschaft (DFG, German Research Foundation) Project No 274762653. The content of the paper is the sole responsibility of the authors and it does not represent the opinion of the European Commission, and the Commission is not responsible for any use that might be made of information contained. The authors are grateful to Matthias

Nützel (DLR, Germany) for his helpful suggestions on a previous version of the manuscript and to the two anonymous referees who reviewed the paper. The computational resources of the DKRZ (Hamburg, Germany) were essential for developing and testing this new version and are kindly acknowledged.

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
