# Peer review of "ESMValTool v2.0 – Technical overview"

_Geoscientific Model Development, 2019_

## Referee Comment (RC1) · Anonymous Referee #1 · 15 Nov 2019

This document constitutes the revision of the paper "ESMValTool v2.0: Technical overview" by Mattia Righi et. al. with the code "gmd-2019-226". This paper describes the second version of a tool for analyzing the quality of the output data for the Earth System Models participating in the Coupled Model Intercomparison Project. It compares the results obtained with the current version of the tool (ESMValTool 2.0) with the previous version (ESMValTool 1.0) and explains each new functionality implemented and the reasons why the software has been improved.

In general, this paper is a substantial contribution to modelling science and the results are discussed in an appropriate and balanced way. The traceability of results is very clear except for a couple of details that I question later. As for the structuring of the document, I think it is well written. The structure follows a coherent, clear and concise order. In summary, I consider that this paper constitutes a good contribution, more specifically to software quality in the field of climate model development, but I have

some questions ordered by relevance that I would like to be solved before making a decision:

- in section 5.3, you describe a continuous integration server to improve code stability, maintenance, and software quality. The tools used to create a continuous integration server are not open-source tools. They have proprietary licenses with a free plan option in which minimum services are offered. Why have not you chosen to use free software tools to create a continuous integration environment? Jenkins, for example, is a solution with these characteristics to manage a continuous integration server and execute automated tests. Another alternative to static code analysis is "Sonarcloud": a tool that detects bugs, duplication and vulnerabilities on code with the possibility to directly integrate with GitHub. Finally, another thing that has caught my attention is that the branches that make up the Git repository are not explained anywhere in the paper or the attached documentation of the software. This makes it very confusing to select and download a specific version of the software, that is, download the version of the master branch, the development branch, preproduction branch... So, I consider necessary an explanation of this, for example in the "readme.md" file;

- When reading the document, the display of some data is a bit tricky: the information of some data are in tables and the position of them is a little uncomfortable when reading. Table 2 is mentioned on page 9 but does not appear until page 16, it could be on page 13 to facilitate its location. Tables 3 and 4 appear before you mention them. They should be mentioned and then appear in a position as close to where they have been mentioned;

- finally, in this paper, the license of the software tool presented is only mentioned one time. Geosci. Model Dev. is a scientific journal that promotes scientific reproducibility and, therefore, open-source/free software. You could place greater emphasis on the type of software license that ESMValTool has to ensure the

reproducibility of the EMSValTool. Therefore, I think you should highlight the license of the ESMValTool, as well as all the tools used throughout the software life cycle, as I mentioned in the first item and how they can help to improve the reproducibility of the CMIP process.

---

## Referee Comment (RC2) · Anonymous Referee #2 · 8 Dec 2019

This paper amply demonstrates that the the developers of ESMValTool have made very good choices for the path by which the tool has evolved. Significant improvements have been made to extend capabilities (regridding, linking external tools), improve performance and simplify the user experience while expanding the general capabilities of the system.

While v2.0 is not strictly backwards compatible with v1.0, the paper amply justifies those aspects where some users may have to make changes. E.g. stricter CF compliance for input data sets is now required.

The authors have also made excellent choices with regard to distinguishing software engineering standards for core functionality vs user-provided extensions. The responsibility for strong testing requirements for the core lies with professionals and like minded end-users, while still enabling the more general scientific end-user to share

custom diagnostics with the community.

---

## Author Comment (AC1) · 17 Jan 2020

**ESMValTool v2.0 - Technical overview**
*M. Righi et al.*
**Replies to referees comments**

We are grateful to the two referees for reviewing our manuscript and providing insightful comments and suggestions. Please find below a detailed response to each comment: reviewers' comments are marked in blue, authors' reply in black, quoted text from the paper in *"italic red"*.

In addition to the changes to address the reviewers' comments, a few updates have been introduced to the manuscript to document some new features implemented in the latest release of the code. In particular:

- Two new preprocessors are now described in Sect. 4: *detrend* (Sect. 4.4) and *land/sea fraction weighting* (Sect. 4.9);
- A new feature of the task manager which prioritizes the tasks which are listed first in the recipe is now mentioned in Sect. 6;
- Table 1 has been extended to document the settings of the two new preprocessors mentioned above, while some settings have been renamed and/or extended (e.g., for the temporal statistics);
- Table 2 has been updated with a few more derived variables, while some were removed since they calculation can now be performed using the preprocessor;
- Table 3 has been extended with more observational datasets, for which a cmorization script is now available in the ESMValTool;
- Figure 1 has been revised to include the two new preprocessors mentioned above.

**Anonymous referee #1**

In section 5.3, you describe a continuous integration server to improve code stability, maintenance, and software quality. The tools used to create a continuous integration server are not open-source tools. They have proprietary licenses with a free plan option in which minimum services are offered. Why have not you chosen to use free software tools to create a continuous integration environment? Jenkins, for example, is a solution with these characteristics to manage a continuous integration server and execute automated tests. Another alternative to static code analysis is "Sonarcloud": a tool that detects bugs, duplication and vulnerabilities on code with the possibility to directly integrate with GitHub.

We have chosen not to host our own services, because this requires considerable effort to set up and maintain. We use the free services for open source projects provided by CircleCI and Codacy to run open source software. Should these services no longer be free at some point, we could easily move to another service because very little configuration is required and all the tools we use for testing and static code analysis are free and open source.

We have updated section 5.3 to make this clearer: *"CircleCI and Codacy offer free services for open source projects. We use these services to run open source software that could equally easily be run on other infrastructure. On CircleCI the unit tests are run in a Debian Linux docker container with a minimal version of Anaconda pre-installed (https://hub.docker.com/r/continuumio/miniconda3). On Codacy we make use of the various open source Python linters that are bundled into Prospector (https://prospector.readthedocs.io). These tools can also be installed and used on contributors own computers with a minimal effort, as described in our contribution guidelines."*

Finally, another thing that has caught my attention is that the branches that make up the Git repository are not explained anywhere in the paper or the attached documentation of the software. This makes it very confusing to select and download a specific version of the software, that is, download the version of the master branch, the development branch, preproduction branch... So, I consider necessary an explanation of this, for example in the "readme.md" file

The branch structure of the ESMValTool (ESMValCore) was admittedly a bit confusing, so it was revised and simplified. The CONTRIBUTING.md files were updated accordingly. The stable branch is now called *master* in both repositories and the user is always pointed to this branch when accessing the code on GitHub. The same happens when a pull request is submitted: by default, the target branch is always set to *master*. All other branches are feature branches that may at some point be merged, after a pull request is submitted by the corresponding developer and approved by the core development team.

The ESMValTool (ESMValCore) version described in the paper refers to the latest release of the code and is clearly linked in the Code Availability section with a doi to a Zenodo repository, which always points to the latest release. The latest releases can also be retrieved directly from our GitHub repositories, as also mentioned in the Code Availability section.

When reading the document, the display of some data is a bit tricky: the information of some data are in tables and the position of them is a little uncomfortable when reading. Table 2 is mentioned on page 9 but does not appear until page 16, it could be on page 13 to facilitate its location. Tables 3 and 4 appear before you mention them. They should be mentioned and then appear in a position as close to where they have been mentioned.

Thank you for this suggestion. Usually these editorial aspects are taken care of by the production office once the paper is accepted and the final version is generated. We will make sure that all tables are correctly placed during the proof-reading stage.

Finally, in this paper, the license of the software tool presented is only mentioned one time. Geosci. Model Dev. is a scientific journal that promotes scientific reproducibility and, therefore, open-source/free software. You could place greater emphasis on the type of software license that ESMValTool has to ensure the reproducibility of the EMSValTool. Therefore, I think you should highlight the license of the ESMValTool, as well as all the tools used throughout the software life cycle, as I mentioned in the first item and how they can help to improve the reproducibility of the CMIP process.

Licensing aspects and free availability of the used packages is already mentioned in several parts of the papers, but we tried to improve this, putting more stress on these aspects as suggested by the reviewer:

- *"To support the community in this big data challenge, the ESMValTool (Eyring et al., 2016c) has been developed to provide an **open-source**, standardized, community-based software package for the systematic, efficient and well documented analysis of ESM results."* (Introduction)

- *"As for v1.0, ESMValTool v2.0 is released under the **Apache license**. The source code of both ESMValTool and ESMValCore is **freely accessible** on the GitHub repository of the project (https://github.com/ESMValGroup) and is fully based on **freely available** packages and libraries."* (Introduction)

- *"ESMValTool v2.0 is distributed as an **open-source** package containing the diagnostic code and related interfaces"* (Section 2)

- *"Support for other **freely available** programming languages for the diagnostic scripts can be added on request."* (Section 2)

- *"The ESMValTool v2.0 preprocessor is entirely written in Python and takes advantage of the Iris library (v2.2.1) developed by the Met Office (Met Office, 2010-2019). Iris is an **open-source**, community-driven Python 3 package for analyzing and visualizing Earth science data, building upon the rich software stack available in the modern scientific Python ecosystem."* (Section 4).